# Spatial Variability of Grouting Layer of Shield Tunnel and Its Effect on Ground Settlement

**Zhongzheng Wang [1], Dalong Jin [2,3,*] and Chenghua Shi [1]** 

1   School of Civil Engineering, Central South University, Changsha 410083, China;
    zhongzhengwang@csu.edu.cn (Z.W.); csusch@163.com (C.S.)
2   School of Civil Engineering, Beijing Jiaotong University, Beijing 100044, China
3   Department of Civil & Environmental Engineering, National University of Singapore,
    Singapore 119077, Singapore
*   Correspondence: 14115317@bjtu.edu.cn

**Abstract:** This study aims to investigate the effect of the spatial variability of grouting-layer thickness on ground-surface settlement caused by shield tunneling and to provide a rational prediction method. The spatial characteristics of grouting layers were obtained based on statistical analysis. The random finite element method was used to study the effect of spatial variability of different parameters on ground-surface settlement. Simulation results indicate that the spatial variability of the grouting layer has a negative impact on ground settlement. The surface settlement will be underestimated without considering the spatial characteristics of the grouting layer. Thus, a reliable prediction approach of the maximum ground settlement was proposed to control the construction quality.

**Keywords:** random finite element; grouting-layer thickness; ground-surface settlement; statistical analysis

## 1. Introduction

With the development of urban underground spaces, shield-tunneling methods have been widely used in urban subway engineering. However, the excavation of shield tunnels will inevitably cause ground settlement. This sometimes even leads to ground-collapse accidents. Therefore, the prediction of ground settlement caused by shield tunneling plays an important role in controlling construction quality.

During the past years, tunneling-induced ground settlement problems have attracted many researchers. The methods used to analyze this problem can be roughly classified into four categories: empirical formula method, laboratory tests, numeric simulation method and analytical method. For the (a) empirical formula method, Peck [1] derived the empirical formula for predicting the transverse settlement troughs of ground based on a large amount of field data; (b) For laboratory tests, in order to study the effect of different factors on the ground settlement, a series of small-scale models have been tested in the laboratory [2–8]; (c) for numeric simulation, it is common to simulate ground movement by using the finite element method (FEM) [9–14]. In practice, the mechanical properties of the soil have great spatial variability. In order to study the effect of spatial variability on the tunneling-induced ground movement, some numeric investigations with random finite element analysis (RFEA) have also been performed [15,16], providing more reliable references for tunnel construction. For the (d) analytical method, Rowe et al. [17] defined the concept of excavation gap parameter $g$ and proposed a theoretical prediction model for predicting the ground settlement caused by shield tunneling in soft ground. On this basis, Loganathan et al. [18] further studied the ground loss rate caused by tunnel excavation and developed an analytical solution which can consider the oval-shaped deformation of the tunnel. Most of the above studies assumed that the settlement induced by tunnel boring machine

(TBM) is symmetrical or in regular shape. Indeed, fine workmanship usually produces a symmetrical and normal settlement distribution.

Due to the gap between the shield tail and segmental lining, the ground settlement mainly occurs at the shield tail passing stage [19]. Backfill-grouting is critical for filling the gap and reducing ground settlement. However, due to the complexity and uncertainty of geological and environmental conditions, the shield tail gap cannot be completely refilled. Ground settlement after the shield-tail passing can be found even after sufficient grouting material is injected. This is considered to be related to the uneven distribution of the grouting layer and that some injected grouting materials are wasted [20]. Nevertheless, there is still a lack of research on the spatial variability of the grouting layer and its effect on ground settlement.

In this study, the spatial variability of the thickness of the grouting layer in shield tunnel is summarized, first based on collected test data. Based on the Mohr–Coulomb theory, a random finite element model of tunnel excavation is established in a spatial heterogeneous stratum. Two dimensional random variables were used to describe soil parameters and grouting-layer thickness. A Monte Carlo simulation was carried out to analyze the influences of spatial variability of soil parameters and grouting-layer thickness on ground settlement. Finally—to provide a more reliable prediction of the ground settlement caused by shield tunneling—a practical and realistic approach to estimate the tunneling-induced ground settlement is proposed based on the probabilistic method.

## 2. Spatial Variability of Grouting Layers

The shield tunneling method allows excavation while preventing tunnel collapse by assembling steel or concrete segmental blocks in a ring conformation to create a tunnel structure. In order to permit the advance of the shield machine and assemble the segments under the protection of the shield, the excavation diameter is usually larger than the external diameter of the segmental lining, thus a gap exists between the excavation boundary and segmental linings. The purpose of backfill grout injection is to immediately fill voids left by the shield machine body between the tunnel wall and the segmented blocks, preventing subsidence of surrounding ground and adjacent structures. Nevertheless, the distribution of the backfill-grouting cannot be uniformly distributed around segmental linings. As shown in Figure 1, the grouting layer in reality has spatial variability due to the geological randomness and mortar arbitrary flowing. Although the injected grouting is enough to fill the voids, a settlement can immediately occur after the passing of the shield tail. The settlement patterns and amount are considered to be closely related to the spatial variability of grouting layers [21]. The distribution of the grouting layer can be obtained by two methods, ground penetrating radar (GPR) and large visualized model test. GPR is a rapid nondestructive method that can recognize the interfaces of different objects. This method has been widely applied in the detection of tunnel defects to evaluate the quality of backfill-grouting layers [22,23]. The visualized model test is another method which can directly observe the characteristics of grouting layers, and many relevant studies can be found in the literature [24].

Based on previous studies, the statistical characteristics obtained from some monitoring results of the grouting layers are listed in Table 1. The obtained results indicate that the backfill-grouting layers have significant variability. The coefficient of variation (COV) ranges from 0.1 to 0.4. Figure 2 presents the distribution of a grouting-layer thickness; the data are collected from a monitoring section reported by Ding et al. [24]. To further test the normality hypotheses, the Anderson–Darling test within the Matlab was used. The Anderson–Darling test results indicate that the thickness of the grouting layer can be successfully fitted by the normal distribution, passing the test at 5% significance level.

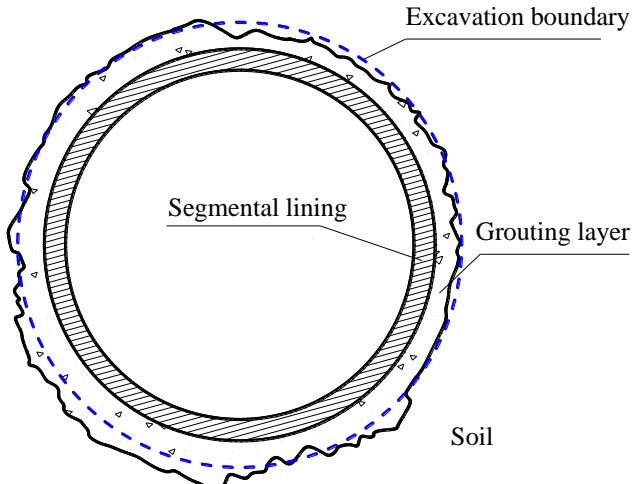

**Figure 1.** Schematic diagram of backfill-grouting-layer distribution.

**Table 1.** Statistic parameters of grouting-layer thickness.

| Source. | Mean | Span | COV | Test Method |
|---------|------|------|-----|-------------|
| Yu et al. (2016) [23] | 300 mm | 220 mm–380 mm | 0.1–0.20 | Ground penetrating radar (GPR) |
| Ding et al. (2019) [24] | 169 mm | 105 mm–240 mm | 0.15–0.40 | Large visualized model test |

Note: COV refers to the coefficient of variation.

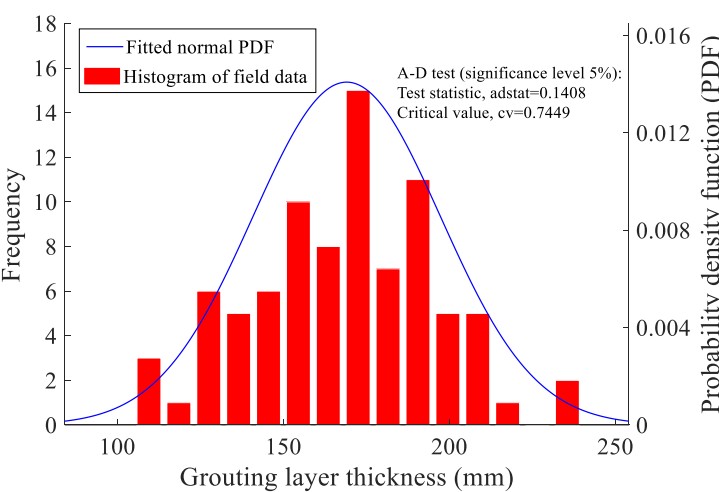

**Figure 2.** Distribution of the grouting-layer thickness.

It has been found that most of the ground settlement occurs at the passing of the shield tail. The distribution of the backfill-grouting layer greatly affects the convergence patterns along the excavation surface of tunnels. As shown in Figure 3, the displacement convergence patterns can be concluded as three types based on different grouting-layer distribution. In Type I, a uniform convergence pattern corresponds to a uniform distribution of the grouting layer. In Type II, the maximum convergence pattern corresponds to as that most of the grouting fluid flows to the bottom of the tunnel. The distribution of the grouting layer can be simplified as a tangent circle outside the tunnel circle. In Type III, the spatial variability convergence pattern corresponds to a grouting layer with random defects. The first two types have been widely used to estimate ground displacement and can be found in many studies [25,26]. However, the ground movement caused by the defective grouting layer is rarely discussed.

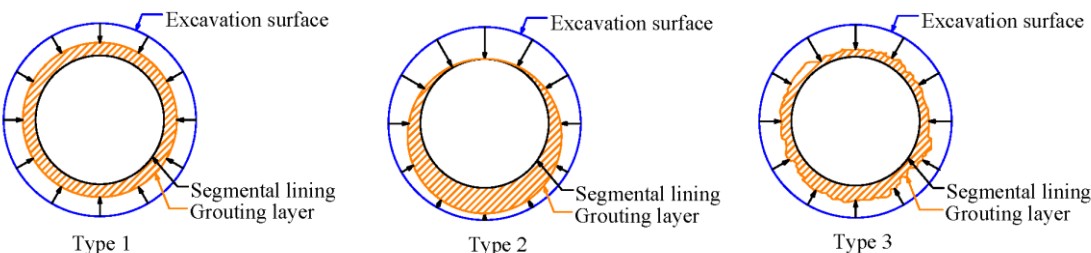

**Figure 3.** Displacement convergence patterns for different grouting-layer distributions.

## 3. Modeling Aspects

### 3.1. FEM Model

In this study, a finite element analysis (FEA) was conducted to discuss the ground settlement induced by shield tunneling considering the backfill-grouting distribution. As shown in Figure 4, a two-dimensional nonlinear plane strain FEA established via using ABAQUS 6.14 program. The model dimension of 40 m (length) × 20 m (depth) was chosen to reduce the influence of boundary conditions on tunneling-induced stress and strain. The constitutive model of soil was based on the Mohr–Coulomb model that included soil bulk density $\gamma_s$, Poisson's radio $v$, friction angle $c$, cohesion force $\varphi$, Young's modulus $E$. The normal displacement of the side boundaries and the bottom boundary was fixed. The whole model adopted the four-node bilinear plane strain quadrilateral element(CPE4). The average mesh size of the stratum was 0.5 m, while the average mesh size of the tunneling area was 0.4 m. The model consisted of 3552 elements and 3675 nodes. The excavation parameters of the tunnel and soil parameters were referred to Shenzhen Metro [27]. The excavation diameter of the tunnel $D$ was 6.0 m, the outer diameter of the lining $D_l$ was 5.6 m, and the burial depth of the tunnel $B$ was 9.0 m. The excavation gap $g$ between the shield tail and lining was 200 mm. The variability of soil properties has been extensively studied in past years [28–36]. Phoon et al. evaluated the geotechnical variability in detail for different types of soil [34]. Based on extensive previous studies, the COVs of soil mechanical parameters $c$, $\varphi$ and $E$ showed significant differences, while the scale of fluctuation (SOF) of soil parameters showed a certain degree of consistency [35,36]. The reference case of this study was to simulate the clay formation. The selection of soil parameters can refer to the above research. Previous studies showed that there is low correlation between random variables of soil parameters, so it is not necessary to consider their correlation. The specific parameters of the reference case are shown in Table 2.

It is not easy to express the thickness of grouting layer in the numeric simulation software. Generally, the excavation gap parameter $g$ is a fixed value. Therefore, this study presents an appropriate simulation method of the spatial variability of grouting layer via randomizing the convergence displacement of tunnel-excavation boundary surface. $\delta$ is equivalent to a mapping of $t$, in which $\delta$ exhibits the same random properties as $t$ (the same COV). Therefore, when modeling random fields, only the randomization process of $\delta$ needs to be considered. When the grouting-layer distribution is determined, the contracted displacement around the tunnel boundary thus can be obtained,

$$\delta = g - t \tag{1}$$

where $\delta$ refers to the radial displacement at an arbitrary position around the tunnel, $g$ refers to the voids between the excavation surface and segmental lining, $t$ refers to the thickness of the grouting layer.

A ground contraction method was adopted in this paper to simulate the soil disturbance after the tunnel excavation. To control the convergence displacement of the excavation surface, the circumference of the tunnel was divided into 48 elements and 48 nodes. The constraint effect of tunnel supporting structure was replaced by a constraint displacement field, in which the displacement constraints were applied to 48 nodes on the circumference of the tunnel.

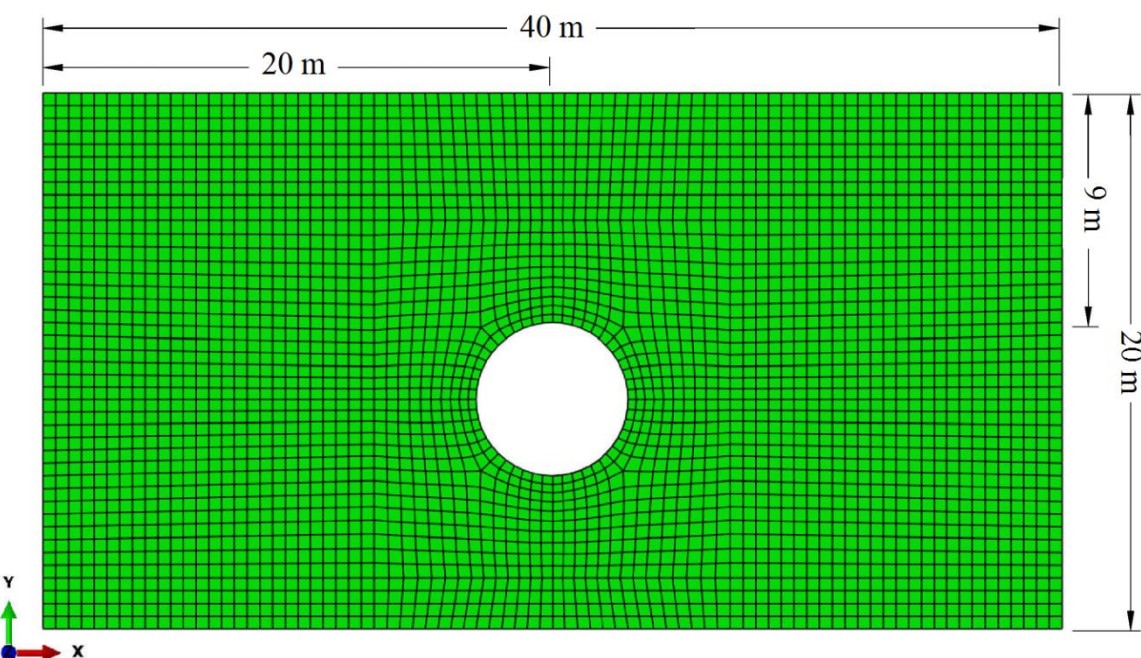

**Figure 4.** Model geometry and mesh gird.

**Table 2.** Input parameters for the reference case.

| Deterministic Variables | Value | Input Variables | Mean Value ($\mu$) | COV (%) | SOF (m) | |
|---|---|---|---|---|---|---|
| | | | | | $s_h$ | $s_v$ |
| $D_l$ (m) | 5.6 | $c$ (kPa) | 10 | 0.20 | | |
| $D$ (m) | 6.0 | $\varphi$ (°) | 25 | 0.10 | 10 | 2.0 |
| $g$ (mm) | 200 | $E$ (MPa) | 20 | 0.15 | | |
| $\gamma_s$ (kN/m³) | 20 | $\delta$ (mm) | 20 | 0.20 | | |
| $v$ | 0.2 | $t$ (mm) | 180 | | – | |

Note: SOF refers to the scale of fluctuation.

In ABAQUS, the value of the constraint field in the polar coordinate system was $\delta$ (negative value indicates convergence to the center of the circle, Figure 5).

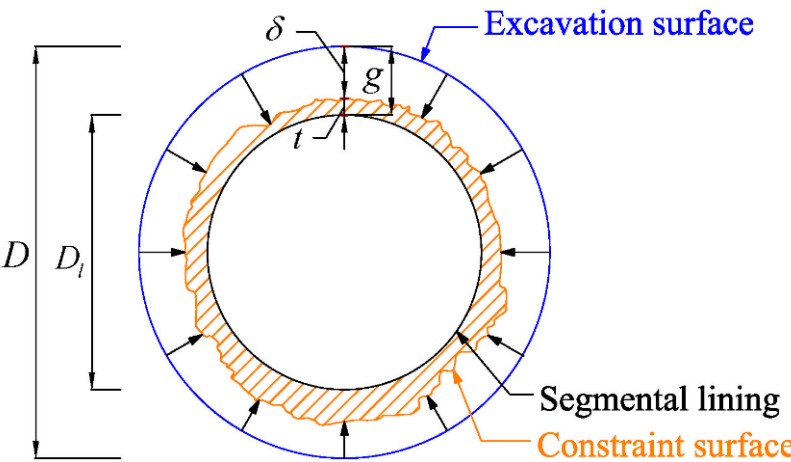

**Figure 5.** Displacement-constrained field.

In order to reflect the function of supporting structure in the tunneling process, the earth stress balance was first carried out by setting the gravity stress of the highest and lowest faces. After that, the constraint displacement field was applied at the same time as the tunnel excavation, which meant that the excavation face contract immediately after the tunnel excavation. The preliminary study showed that the mesh size and the distance from the tunnel to the side and bottom boundaries were sufficient to produce convergence under the reference case.

*3.2. Random Model*

It is well known that ground-surface settlement is closely related to soil properties. When the Mohr–Coulomb model was adopted, the main properties of soil are $E$, $c$, $\varphi$ and $v$. Both practice and theory showed that the spatial variability of Young's modulus $E$ has a significant impact on the ground-surface settlements. When RFEM was used to simulate the spatial variation of soil shear strength, the spatial variability and interrelationship of soil properties ($c$ and $\varphi$) also affected the surface settlement [35]. In addition, Fenton et al. pointed out that the Poisson's ratio $v$ of soil has smaller relative spatial variability and its importance to settlement statistics is only second–order [37]. Thus, the Poisson's ratio in this study is fixed at 0.2 over the entire soil mass for all simulations.

The spatial variability of soil has three indices: (a) Mean of the soil parameters; (b) Coefficient of variation, which describes the variability of the soil parameters; (c) Scale of fluctuation, which describes the spatial correlation. The SOF is loosely defined as the longest distance within which a similarity between two points can be observed. Small values of SOF imply that the correlation function falls off rapidly to zero (i.e., the correlation between two points becomes rapidly smaller), which leads to rougher random fields. On the other hand, for increasing values of SOF the soil property field becomes smoother, in other words, the field shows less variability converging to a uniform field. Many scholars had done much research on the distribution range of COV and SOFs of different soils. This study intends to take the clay soil, a representative soil, as the research object. A normally distributed random field was adopted in this study to represent the soil properties and the grouting layer. The quadratic exponential autocorrelation structure of a two-dimensional random field is mathematically expressed as,

$$\rho = \exp\left\{-\left[\left(\frac{x_i - x_j}{s_h}\right)^2 + \left(\frac{y_i - y_j}{s_v}\right)^2\right]\right\} \tag{2}$$

where $(x_i, y_i)$ and $(x_j, y_j)$ define the spatial center coordinates of two arbitrary contacts.

A standard normally distributed random field can be approximately represented by an M-term K–L expansion,

$$G_s(x) \approx \sum_{i=1}^{M} \sqrt{\lambda_i} \xi_i \psi_i(x) \tag{3}$$

where $\xi_i$ are independent standard normal variables, $M$ is the truncation term of the expansion, $\lambda_i$ and $\psi_i(x)$ denote the eigenvalues and eigenfunctions of the autocorrelation function in Equation (2). x denotes a spatial vector of the random process. The convergence and the accuracy of the K–L expansion depend on the number of the term $M$, which relates to the domain of the random field and the scale of fluctuation. The larger the random field domain and the smaller the autocorrelation distance, the more the terms are required in Fourier series to achieve a given accuracy.

The normal random fields of the soil properties or grouting layer can be expressed as,

$$G_s(x) = \mu + \sigma G(x) \tag{4}$$

where $\mu$ and $\sigma$ are the mean and standard deviation of the grouting-layer thickness or soil properties.

The variability of soil properties has been extensively studied in past years. Phoon et al. proposed the recommended values of geotechnical variability for different types of soil [34]. Based on the suggested value of COVs, an interval of 0.1~0.3 can cover some typical soils. Some studies assumed that $s_h$ and $s_v$ are the same, which was to presume that the soil is an isotropic correlation structure. However, the variability of natural soil is usually anisotropic in reality, and the horizontal SOF $s_h$ is greater than the vertical one $s_v$ for typical soil stratum [36]. Jiang et al. summarized several previously published articles and recommended that the horizontal scale of fluctuation ranges between 10 m and 40 m, and the vertical scale of fluctuation ranges between 0.5 and 3.0 m [31]. The recommended range can be adopted in this work.

The focus of this study was to explore the influence of randomness of grouting-layer thickness on the ground-surface settlement. Under the assumption that the $g$ is fixed, the randomness of grouting thickness would lead to the randomness of excavation boundary convergence. That is, the random distribution of radial displacement of each node reflected the spatial variability of grouting-layer thickness. The randomness of the grouting-layer thickness was simulated by controlling the displacement of each node on the excavation boundary. Regarding the randomness of grouting-layer thickness, the COV of the grouting layer ranges from 0.1 to 0.4 (see Section 2 for details). The SOF of the grouting layer is difficult to determine statistically due to the limited collected data. However, it is recognized the spatial fluctuation of the grouting layer in practice mainly relies on the grouting pipe layout. Thus, the SOF of the grouting layer is assumed as the circumferential spacing of the grouting pipes. In this study, a one-dimensional random field is employed and the SOF of the grouting layer is taken as 4.71 m. This SOF corresponds to a four grouting pipes distribution, which is the most commonly adopted in tunnel projects.

In short, parametric studies were divided into two categories. In (a) traditional soil parameters, the COV of Young's modulus, COV of cohesion, COV of friction angle, SOF of the horizontal direction and SOF of the vertical direction, and (b) grouting-layer thickness: the mean value of grouting-layer thickness $t$ (Corresponding to the gap parameter $\delta$), COV of grouting-layer thickness. In the random analysis of each parameter, Monte Carlo simulations with 300 realizations were conducted to obtain a statistically stable result, with an error of less than 0.2% (Figure 6). All random variables were considered as truncated normal distribution in order to eliminate some unrealistic data using statistical intervals of the input parameters. The details of the random parameters are shown in Table 3.

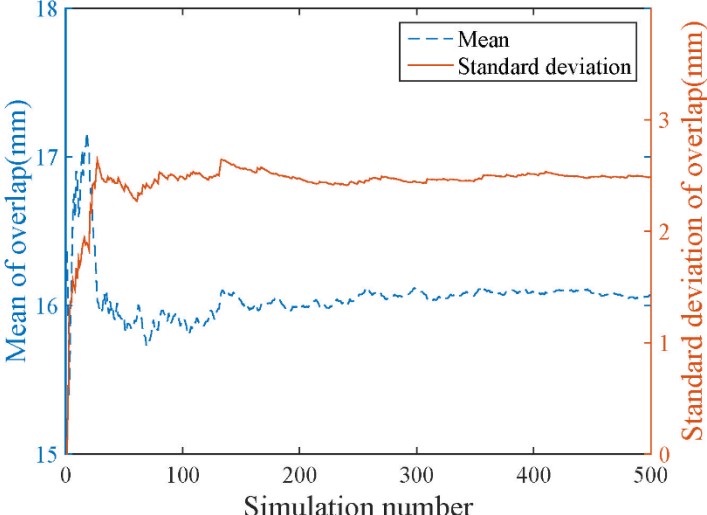

**Figure 6.** Validation of mean and standard deviation.

**Table 3.** Reference case and parametric study.

| Remarks | COV (*c*) | COV (*φ*) | COV (*E*) | SOF (soil)/m | | *t* (mm) | *δ (mm)* | COV (*t*) |
| --- | --- | --- | --- | --- | --- | --- | --- | --- |
| | | | | $s_v$ | $s_h$ | | | |
| Reference case | 0.2 | 0.1 | 0.15 | 2.0 | 10 | 180 | 20 | 0.2 |
| Effect of COV of soil cohesion | 0.1 * | 0.1 | 0.15 | 2.0 | 10 | 180 | 20 | 0.2 |
| | 0.2 * | 0.1 | 0.15 | 2.0 | 10 | 180 | 20 | 0.2 |
| | 0.3 * | 0.1 | 0.15 | 2.0 | 10 | 180 | 20 | 0.2 |
| Effect of COV of soil friction angel | 0.2 | 0.1 * | 0.15 | 2.0 | 10 | 180 | 20 | 0.2 |
| | 0.2 | 0.2 * | 0.15 | 2.0 | 10 | 180 | 20 | 0.2 |
| | 0.2 | 0.3 * | 0.15 | 2.0 | 10 | 180 | 20 | 0.2 |
| Effect of COV of soil Young's modulus | 0.2 | 0.1 | 0.05 * | 2.0 | 10 | 180 | 20 | 0.2 |
| | 0.2 | 0.1 | 0.15 * | 2.0 | 10 | 180 | 20 | 0.2 |
| | 0.2 | 0.1 | 0.25 * | 2.0 | 10 | 180 | 20 | 0.2 |
| Effect of soil SOF | 0.2 | 0.1 | 0.15 | 2.0 * | 10 | 180 | 20 | 0.2 |
| | 0.2 | 0.1 | 0.15 | 2.0 * | 10 | 180 | 20 | 0.2 |
| | 0.2 | 0.1 | 0.15 | 3.0 * | 10 | 180 | 20 | 0.2 |
| | 0.2 | 0.1 | 0.15 | 2.0 | 10 * | 180 | 20 | 0.2 |
| | 0.2 | 0.1 | 0.15 | 2.0 | 20 * | 180 | 20 | 0.2 |
| | 0.2 | 0.1 | 0.15 | 2.0 | 30 * | 180 | 20 | 0.2 |
| Effect of grouting-layer thickness | 0.2 | 0.1 | 0.15 | 2.0 | 10 | 190 * | 10 * | 0.2 |
| | 0.2 | 0.1 | 0.15 | 2.0 | 10 | 180 * | 20 * | 0.2 |
| | 0.2 | 0.1 | 0.15 | 2.0 | 10 | 170 * | 30 * | 0.2 |
| Effect of COV of grouting layer | 0.2 | 0.1 | 0.15 | 2.0 | 10 | 180 | 20 | 0.1 * |
| | 0.2 | 0.1 | 0.15 | 2.0 | 10 | 180 | 20 | 0.2 * |
| | 0.2 | 0.1 | 0.15 | 2.0 | 10 | 180 | 20 | 0.3 * |
| | 0.2 | 0.1 | 0.15 | 2.0 | 10 | 180 | 20 | 0.4 * |

Note: "*" refer to association of variables.

## 4. Results and Analysis

The maximum surface settlements in the random models were extracted and the hypothesis test was conducted. Figure 7 presents the empirical cumulative distribution function (CDF) of the maximum surface settlement. The Anderson–Darling test results indicate that the overall settlements of all the cases can be successfully fitted by the normal distribution, passing the test at the 5% significance level. After confirming that the maximum surface settlement follows the normal distribution, the probability density function (PDF) of the maximum settlements was plotted. The probability density function describes the distribution of a variable. The larger the peak point of the PDF curve, the larger the mean of the maximum surface settlement. The flatter the PDF curve, the stronger the variability of the maximum surface settlement.

Figure 8 shows the effect of the COVs of soil on the maximum surface settlement. It can be seen that the COV of the soil strength parameters, friction angle and cohesion, has little effect on the PDF of the maximum settlement. While the COV of the soil Young's modulus has a great influence on the maximum settlement. The maximum settlement PDF becomes taller and thinner with the decrease of the COV of the soil Young's modulus, indicating the variability of Young's modulus is the main geological reason for the fluctuation of the surface settlement. Figure 9 presents the effect of the SOFs of soil on the maximum surface settlement. It shows that when the horizontal SOF ($s_h$) increases from 10 m to 30 m, the mean value of the maximum surface settlement increases slightly. The vertical SOF ($s_v$) nearly does not influence the PDF of surface settlement. This is because the vertical SOF ($s_v$) is relatively smaller than the soil cover and the surface settlement is a result of stress and strain propagation in the global space.

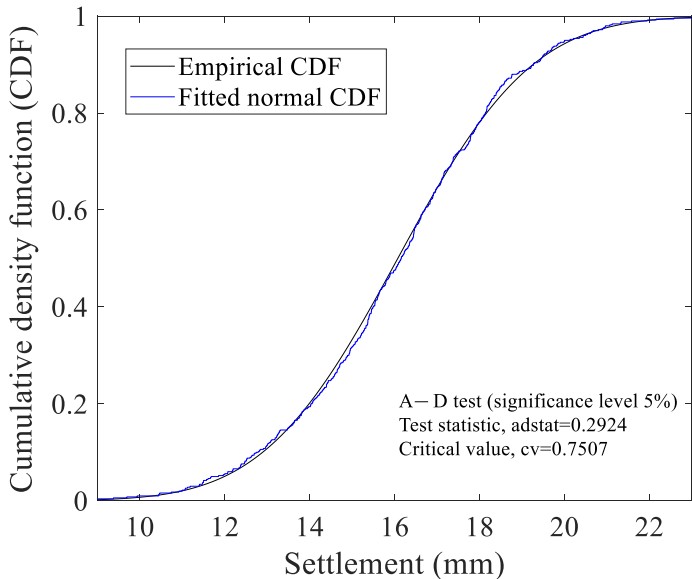

**Figure 7.** Anderson–Darling goodness-of-fit test of normal distribution.

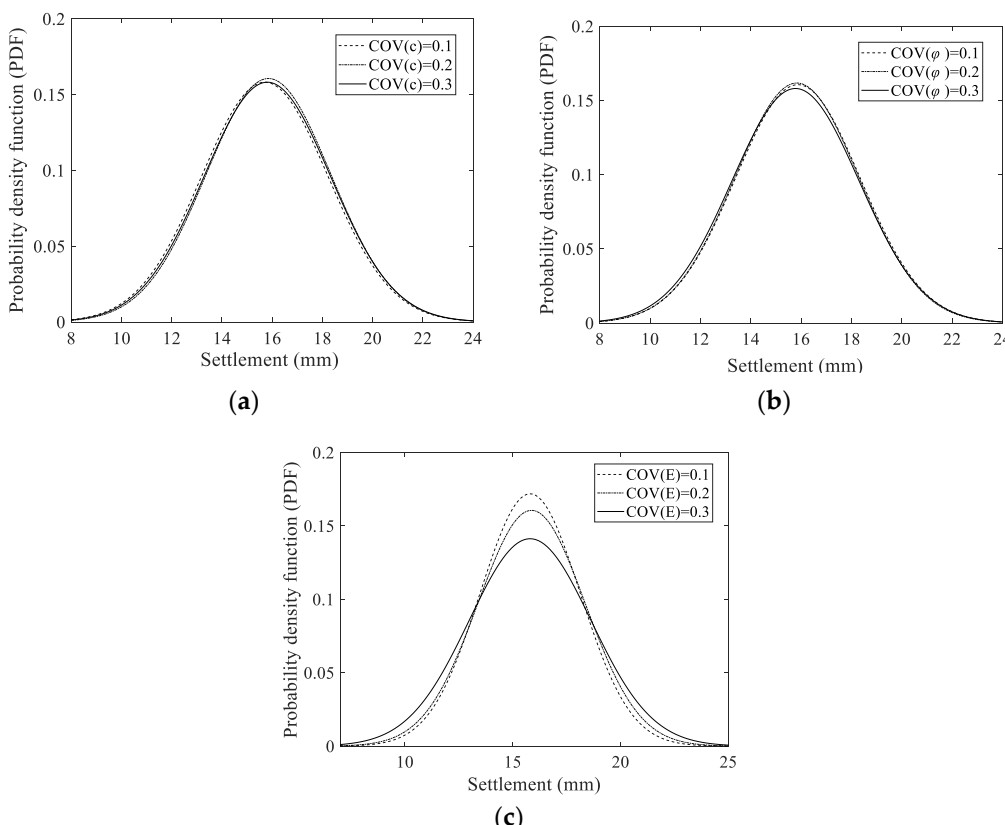

**Figure 8.** Effect of the soil COVs on settlement. (**a**) COV(*c*); (**b**) COV(*φ*); (**c**) COV(*E*).

Figure 10a shows that when the gap parameter $\delta$ increases from 10 mm to 30 mm (corresponding to the grouting-layer thickness *t* from 190 mm to 170 mm), the mean value and standard deviation of the maximum ground-surface settlement both increase appreciably. The apparent increase in the mean is well understood, which is mainly because the larger the radial gap, the greater the amplitude of the ground movement. It is remarkable that the standard deviation also rises observably, indicating

that the increase of the gap parameter $\delta$ can also increase the variability of the ground settlement. This failure risk induced by the variability of the settlement should be fully evaluated. Figure 10b demonstrates that when COV($t$) of the grouting thickness increases from 0.1 m to 0.4, the mean value changes little, whereas the standard deviation rises dramatically. It means that the random defects in the grouting layer can cause a larger maximum ground-surface settlement.

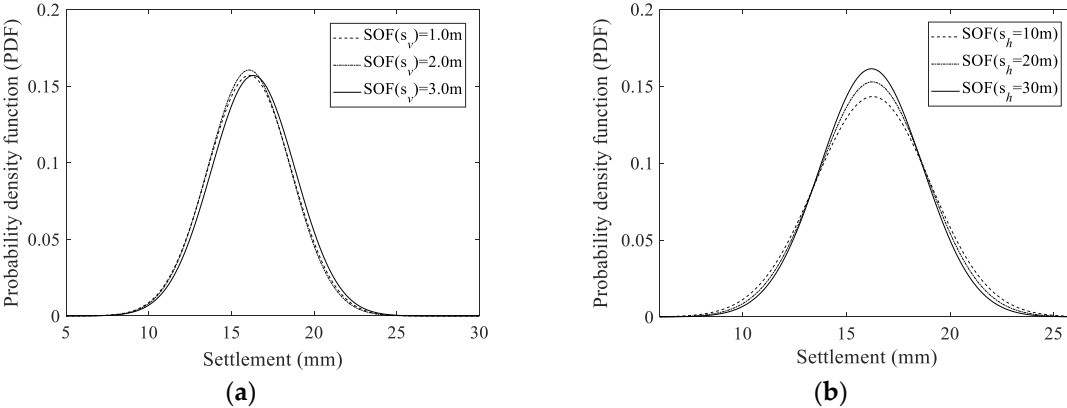

**Figure 9.** Effect of the soil SOFs on settlement. (**a**) $s_v$; (**b**) $s_h$.

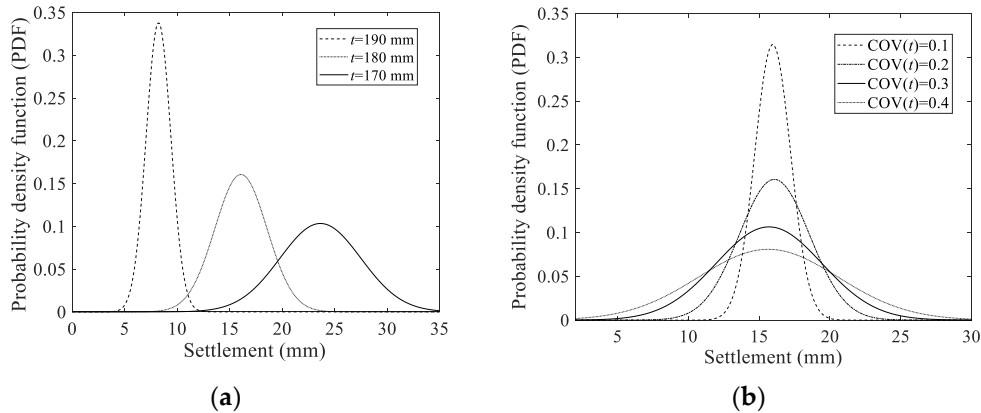

**Figure 10.** Effect of the grouting-layer parameters on settlement. (**a**) Grouting-layer thickness; (**b**) COV($t$).

## 5. Engineering Applications

The construction of a tunnel needs to strictly control the ground-surface settlement, especially in urban metro projects. The above study shows that when the spatial variability of the grouting layer has a significant impact on the ground settlement, which cannot be ignored in the tunneling process. It emphasized the importance of the investigation of the grouting layer quality. The construction safety may not be guaranteed by the amount of the grouting injection volume. Additionally, the spatial variability of soil mechanics parameters is a secondary factor affecting the ground settlement. A detailed site characterization before the tunnel construction should be conducted to obtain enough information for reducing uncertainties in soil parameters.

In the design phase, a "characteristic value" is often used to account for the variability of materials [38]. Eurocode 7 recommends that the "characteristic value of a geotechnical parameter shall be selected as a cautious estimate of the value affecting the occurrence of the limit state" [38]. In the case of this study, it means selecting the 95% fractile as the "characteristic value". Engineers can optimize the construction plan according to the predicted "characteristic ground-surface settlement".

Generally speaking, there are two methods for predicting the maximum ground settlement $s_{max}$ under the random field: (a) The theoretical formula derived from the classification of probability distribution; (b) The prediction curve based on the statistical value of numeric simulation. In this study, the first method is used to provide a practical and realistic approach for maximum settlement estimation.

According to the normal probability distribution function, the maximum ground settlement $s_{max}$ of 95% confidence degree can be obtained,

$$s_{\max} = \mu + 1.65\sigma \tag{5}$$

where $\mu$ is the mean value of a specific random case, $\sigma$ is the standard deviation of a specific random case.

The definition of $COV(t)$ is shown as below,

$$COV(t) = \frac{\sigma_t}{\mu_t} \tag{6}$$

where $\mu_t$ is the mean value of grouting-layer thickness, $\sigma_t$ is the standard deviation of grouting-layer thickness.

The theoretical formula for calculating the $s_{max}$ can be obtained by combining Equations (5) and (6),

$$s_{\max} = \mu_t(1.65 \times COV(t) + 1) \tag{7}$$

In practice, the mean value of the grouting-layer thickness can be easily obtained using the injected grouting volume. Thus, engineers can use this approach to evaluate the maximum settlement once the grouting distribution is determined.

The characteristic value of the numeric simulations is selected as the 95% fractile of the overall simulations to draw the numeric simulation curve. Figure 11 shows the comparison between the maximum settlement obtained from the theoretical prediction approach and numeric simulations. It can be seen from this figure that the maximum settlement increase with the increasing COV of grouting-layer thickness. The slope of the predicted curve based on numeric simulation is smaller than the one based on the theoretical formula, whereas small discrepancy is noted with the increase of the COV of grouting layer. The predicted settlement is a little larger than the numeric results, indicating a conservative estimation of the surface settlement. The discrepancy is around 10% when the COV of grouting layer reaches 0.4, which can be accepted in the engineering practice.

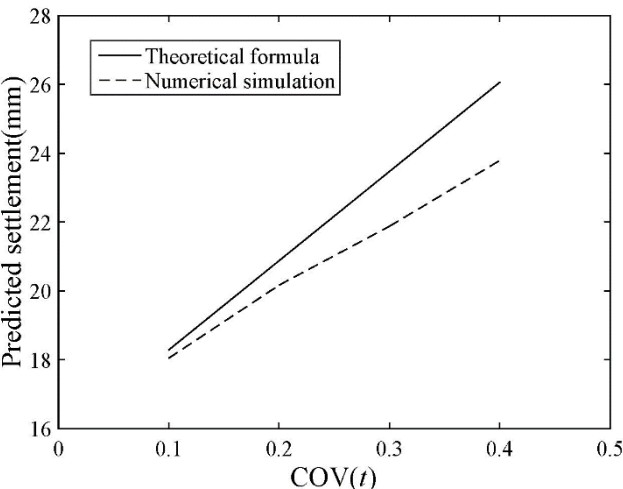

**Figure 11.** Predicted curve for ground-surface settlement.

## 6. Conclusions

In this study, a random finite element model of tunnel excavation was established. The effect of spatial variability of soil parameters and variability of grouting-layer thickness on the ground-surface settlement was studied, respectively. The major conclusions are summarized as follows:

The grouting layer of a shield tunnel shows a great spatial randomness along the circumferential direction. According to the statistical analysis, the thickness of the grouting layer follows a normal distribution and its coefficient of variation ranges from 0.1 to 0.4.

In terms of soil parameters, the COV of the soil strength parameters, friction Angle and cohesion, has little effect on the PDF of the maximum surface settlement, while the COV of the soil Young's modulus shows highly sensitive to the maximum surface settlement. Both the horizontal SOF ($s_h$) and the vertical SOF ($s_v$) nearly have no influence on the PDF of maximum surface settlement.

The spatial variability of the grouting layer shows a significant influence on the ground settlement. The mean value and standard deviation of maximum surface settlement increase with the decrease of the grouting-layer thickness. The ground settlement will be underestimated without considering the spatial variability of the grouting layer of the shield tunnel.

In order to consider the influence of the spatial randomness of grouting layers, a practical prediction approach for ground settlement was given based on 95% fractile. It is worth noting that the value predicted by the theoretical formula is a little higher than the results from the Random Finite Element method.

**Author Contributions:** D.J. proposed the idea of the paper, Z.W. conducted the numerical simulations, and C.S. revised the paper. All authors have read and agreed to the published version of the manuscript.

**Funding:** This work was supported by the National Natural Science Foundation of China (Project NO. 51778636). This work was also funded by the National Defense Science and Technology Innovation Special Zone.

**Acknowledgments:** This work was supported by the National Natural Science Foundation of China (Project NO. 51778636).

**Conflicts of Interest:** The authors declare no conflict of interest.

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
