# Peer review of "Spatial Variability of Grouting Layer of Shield Tunnel and Its Effect on Ground Settlement"

_applsci, doi:10.3390/app10145002_

Round 1
Reviewer 1 Report
The reviewer confirmed the need for the subject, the orderliness of its compositional logic, and the beauty of the chart in this paper.
Therefore, the reviewer will judge it as accepted.
Author Response
Thanks for the reviewer's affirmation.
Reviewer 2 Report
Normally the figures and tables appear in the same sequence as they are located in the text, which is not the case with figure 2 and table 1, figure 6 and table 3.
It is necessary to standardize bibliographic references, because there are parts of the text in which they appear by numbers and others by authors, year.
Author Response
Piont 1:Normally the figures and tables appear in the same sequence as they are located in the text, which is not the case with figure 2 and table 1, figure 6 and table 3.
Response 1:I have swapped Figure 2 for Table 1, Figure 6 for Table 3.
Piont 2:It is necessary to standardize bibliographic references, because there are parts of the text in which they appear by numbers and others by authors, year.
Response 2:I've standardized the bibliography references, and it's all digitally displayed.
Reviewer 3 Report
This paper has investigated the influence of the spatial variability of the soil geometric parameters and grouting layer thickness on the ground surface settlement caused by shield tunneling in order to provide a reliable prediction method. Firstly, the spatial properties of the grouting layer have been generated by statistical method. Secondly, the so-called random finite element method (RFEM) has been used to implement the effect of spatial variability of various parameters ont he ground surface settlement. Giving as a concluding result, this has significant negative impact on the ground settlement which is underestimated.
The topic undertaken in this paper is very important from the viewpoint of engineering, but from mathematical point of view, the paper uses conventional methods. Thus, the work can be considered as a very good engineering work rather than a highly ranked scientific or research work. In view of this, the submitted paper should be rejeceted, however, the topic can include interesting points and aspects for the readers of this issue. So I can recommend for publication in the Applied Sciences, but the peresnt form of the article would require at least a minor revision, concerning a comprehensive grammatic correction.
Author Response
Point 1: I can recommend for publication in the Applied Sciences, but the peresnt form of the article would require at least a minor revision, concerning a comprehensive grammatic correction.
Response 1: I have completed a comprehensive grammatic correction.